# Turbid Coral Reefs: Past, Present and Future—A Review

**Adi Zweifler (Zvifler)** [1,2,*], **Michael O'Leary** [1], **Kyle Morgan** [3] and **Nicola K. Browne** [4]

1   School of Earth Sciences, University of Western Australia, Perth, WA 6009, Australia;
    mick.oleary@uwa.edu.au
2   Oceans Institute, University of Western Australia, Perth, WA 6009, Australia
3   Asian School of the Environment, Nanyang Technological University, Singapore 308232, Singapore;
    kmorgan@ntu.edu.sg
4   School of Molecular and Life Sciences, Curtin University, Perth, WA 6102, Australia;
    nicola.browne@curtin.edu.au
*   Correspondence: adizweifler@gmail.com

**Abstract:** Increasing evidence suggests that coral reefs exposed to elevated turbidity may be more resilient to climate change impacts and serve as an important conservation hotspot. However, logistical difficulties in studying turbid environments have led to poor representation of these reef types within the scientific literature, with studies using different methods and definitions to characterize turbid reefs. Here we review the geological origins and growth histories of turbid reefs from the Holocene (past), their current ecological and environmental states (present), and their potential responses and resilience to increasing local and global pressures (future). We classify turbid reefs using new descriptors based on their turbidity regime (persistent, fluctuating, transitional) and sources of sediment input (natural versus anthropogenic). Further, by comparing the composition, function and resilience of two of the most studied turbid reefs, Paluma Shoals Reef Complex, Australia (natural turbidity) and Singapore reefs (anthropogenic turbidity), we found them to be two distinct types of turbid reefs with different conservation status. As the geographic range of turbid reefs is expected to increase due to local and global stressors, improving our understanding of their responses to environmental change will be central to global coral reef conservation efforts.

**Keywords:** turbidity; coral reef; sedimentation; climate change; resilience

## 1. Introduction

Turbidity is a key water quality parameter that represents the amount of light absorbed or scattered in the water column by suspended particulate matter (SPM) [1,2]. SPM is composed of both inorganic material, usually terrestrial sediment delivered through fluvial (riverine) or aeolian (wind-driven) processes and/or resuspended seafloor sediments, as well as dissolved and particulate organic material, such as phytoplankton (measured as chlorophyll a), zooplankton and bacteria [2–5]. As a consequence, turbid reefs are light-limited coral habitats, and are typically situated in shallow coastal water settings (<10 m depth; <20 km from the coast).

Despite occupying 30% of reefs in the Coral Triangle and 12% of reefs globally [6], turbid coral reefs are relatively unexplored. The lack of data on turbid reefs is largely due to logistical issues associated with working in low visibility conditions both directly (in situ) and indirectly using remote sensing technologies [7]. This has resulted in a poor understanding of how these reefs function, from the individual coral to the reef ecosystem. Traditionally, suspended sediments are considered to have negative impacts on coral reefs (e.g., reduced coral energy production, clogged corallites, coral tissue abrasion and/or smothering), reducing coral cover, diversity [8–11] and resistance [3,12,13]. Over the last 20 years, however, several studies have documented high coral cover on turbid reefs [10,14–20] and elevated resilience to prolonged periods of high sea surface temperatures (SSTs) that have caused severe bleaching at nearby clear-water locations [6,7,18,21,22].

Moreover, evidence from the recent geological record demonstrates that coral reefs during the mid- to late-Holocene initiated and accreted despite sustained exposure to turbidity and sedimentation [23–27]. These lines of evidence suggest that turbid coral communities can tolerate natural marginal growth conditions, which may provide greater resilience to both local and global threats, making turbid reefs potentially critical coastal habitats to focus coral reef conservation efforts.

To understand the potential resilience of turbid corals to climate change and local anthropogenic stressors, and to integrate turbid reefs into a more robust conservation framework, we must first define a turbid reef by identifying the lower boundary of turbidity thresholds (i.e., severity, frequency, duration). Unfortunately, there is limited empirical turbidity data with most assessments of turbid reefs based primarily on reef characteristics (e.g., coral cover, complexity [28]). The lack of quantitative thresholds is partly due to the high variability in environmental conditions (e.g., light, temperature, nutrient, pH) these reefs experience over a range of temporal and spatial scales, which are expensive and difficult to capture. As such, perceptions of what is considered to be a turbid reef often depend on the location, the environmental contrast to nearby offshore reefs, and the researchers' own observational experience.

Sources of turbidity (e.g., river runoff, dredging) can be broadly classified as natural or anthropogenic. This distinction could potentially be a useful tool for conservation management. For example, naturally turbid reefs have established and continue to grow under high turbidity conditions [27] where particulate matter is continuously resuspended by wind-driven waves, (e.g., inshore Great Barrier Reef [8,10]), strong tidal currents (e.g., Kimberley, Western Australia [29]) and/or river discharge plumes (e.g., Abrolhos, Brazil [30]). In contrast, anthropogenic turbid reefs (e.g., Singapore reefs [31]) have experienced recent (<70 years) increases in terrigenous sediment delivery due to changes in land use (e.g., coastal development, dredging, catchment deforestation, agriculture) and in sediment resuspension rates due to human activities (e.g., ship traffic, fishing trawlers), alongside climate change-driven increases in rainfall, resulting in greater land runoff (Figure 1) [32–36]. Consequently, many anthropogenic turbid reefs situated nearby urban centers or modified coastal catchments have reduced reef function [37–39] and decreased habitat availability as reefs vertically compress their depth range [40]. As such, these reefs may represent a different reef type (ecology, function and resilience) that requires distinction from natural systems, particularly when assessing their value for reef conservation management.

Here we review the available scientific literature found on the past, present and future of turbid coral reefs. We begin by summarizing current definitions of turbid reefs, and reevaluating their environmental and ecological characteristics (e.g., suspended sediment loads, sediment accumulation rates, community composition and reef matrix), to provide a new classification of turbid reefs based on their sediment exposure regime. The 'past' focuses on the methods currently used to reconstruct paleoecological communities from the geological record of natural turbid reefs, while the 'present' focuses on our current knowledge of turbid coral communities (e.g., spatial distribution and function). To assess if natural and anthropogenic turbid reefs represent distinct reef types, we focus on two well-studied regions; (1) Paluma Shoals Reef Complex (PSRC), a nearshore natural turbid reef complex situated on the central Great Barrier Reef (GBR), Australia and (2) the offshore reefs of southern Singapore, which have been exposed to increasing anthropogenic sediment levels since the city's establishment in 1819 [41]. The 'future' then explores current questions regarding turbid reef expansion and community responses to increasing local (e.g., sediment loads, eutrophication) and global climate change impacts (e.g., rising SSTs, sea-level rise, increased storm severity and ocean acidification). Finally, we highlight potential resilience attributes and current knowledge gaps in our understanding of turbid reefs' response to environmental changes in the Anthropocene.

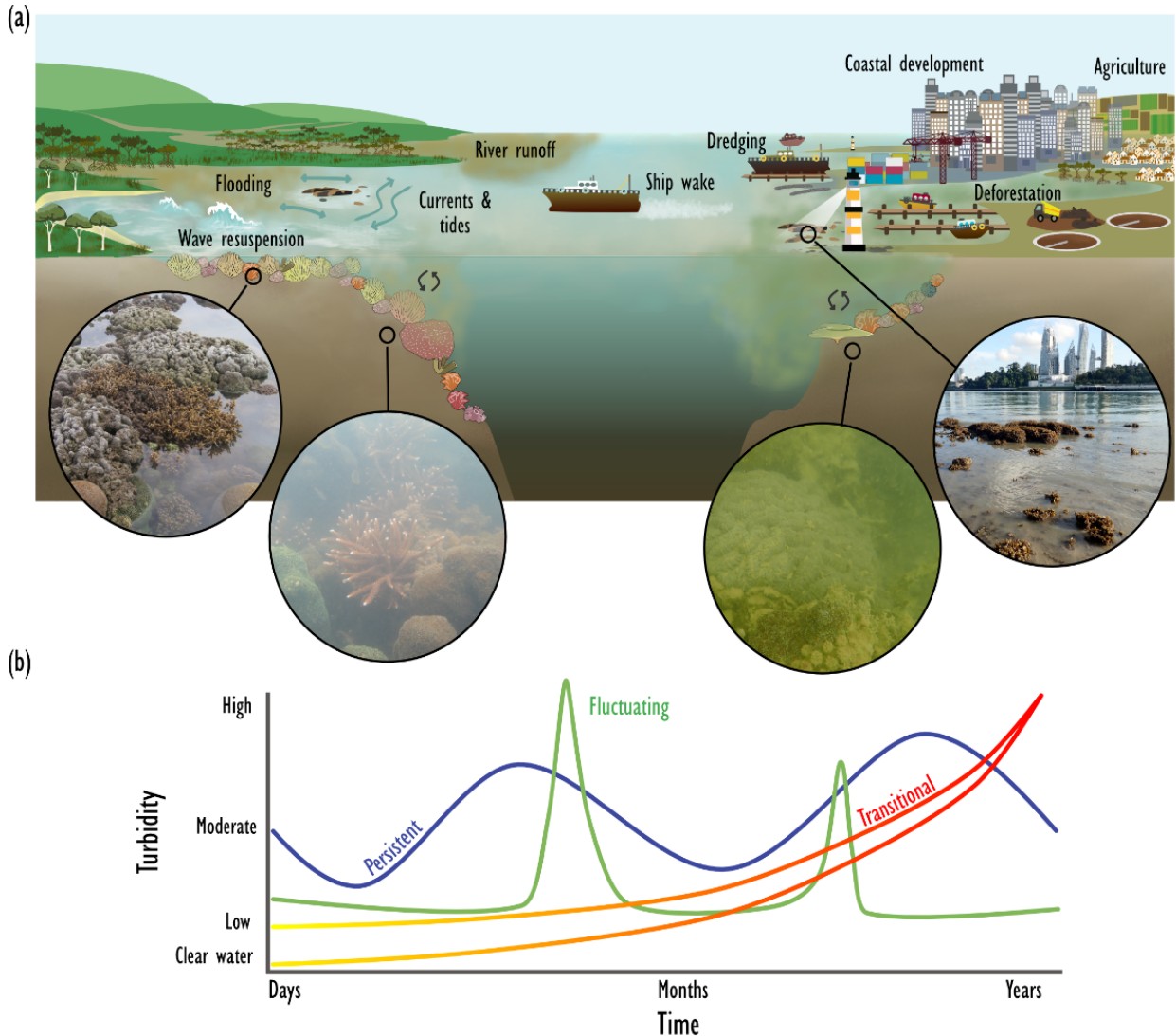

**Figure 1.** (**a**) Natural turbid reef (on the left, two photos of PSRC by Nicola Browne) and VS anthropogenic turbid reef (on the right, two photos of Singapore reefs by Kyle Morgan). (**b**) Types of turbidity regimes on a temporal scale: persistent (blue), fluctuating (green) and transitional (yellow-red). $y$ axis turbidity scale: low (<5 mg L$^{-1}$/<15 NTU), moderate (5 mg L$^{-1}$/15 NTU) and high (>50 mg L$^{-1}$/>150 NTU).

## 2. Methods–Searching for Turbid Reefs (in the Literature)

To gather all known information on turbid reefs (natural and anthropogenic) past and present, a systematic literature review was carried out (Figure 2) [42,43]. The Google Scholar and Web of Science databases were searched using the terms: turbid AND reef AND coral OR coral (larvae OR recruits) AND coral (physiology OR survival OR growth OR resilience) AND reef (ecology OR geology) AND sedimentation (regime OR event). References from review papers [3,9,13,28,31,33,44–55] were also compiled to ensure all relevant papers were acquired. To identify discussions on turbid reefs in the context of coral reef initiation, geological past and future climate change a broader search was manually conducted.

To create a global distribution map of turbid reefs (Figure 3) from this list (*n* = 284), 75 records were excluded for not satisfying our criteria of focused research on coral ecology, geology and/or physiology under turbid conditions. Further, 36 full-text articles were excluded for one of the following reasons: (1) review papers, (2) not location-specific, or (3) artificially ex situ-induced turbidity. For the remaining 173 papers, the following data were recorded:

citation, year, geographic location, study site, turbidity source (natural/anthropogenic) and research discipline (Table S1). Turbid reefs were classified as natural or anthropogenic based on the author's description of the reef (e.g., coral cover, composition), reef setting (e.g., close to urban settlement) and turbidity source (e.g., deforestation, wave-driven). These descriptions were also used to create subcategories within the natural and anthropogenic categories. Natural turbid reefs were divided into: (1) river runoff, or (2) hydrodynamic regime (e.g., tides, currents and wind-driven waves), and anthropogenic turbid reefs were divided into: (1) land use (e.g., agriculture runoff, land reclamation, deforestation), or (2) dredging. Those studies that indicated multiple turbidity sources (e.g., natural and anthropogenic) were classified as mixed (e.g., river runoff/land use).

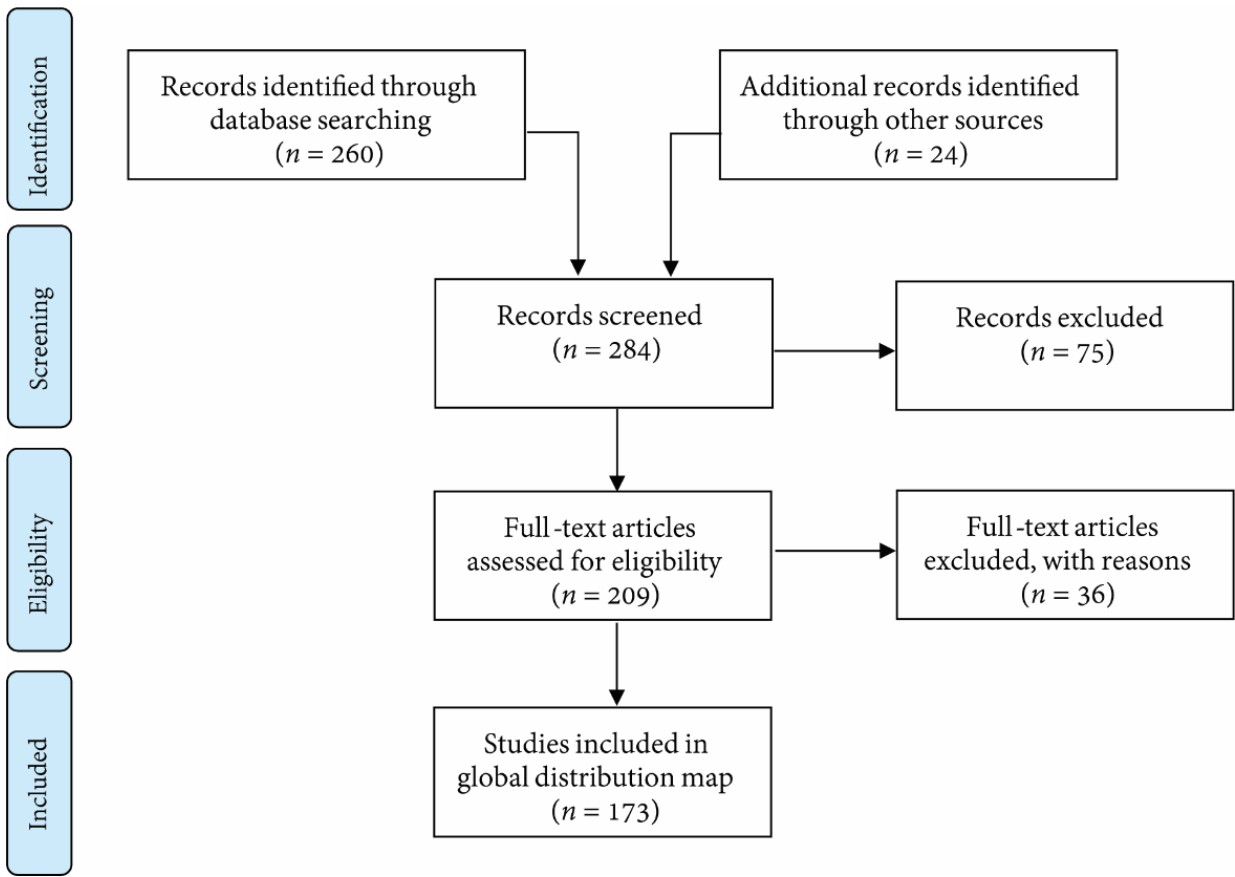

**Figure 2.** PRISMA flowchart of the systematic review papers screening process for creating the turbid reefs global distribution map.

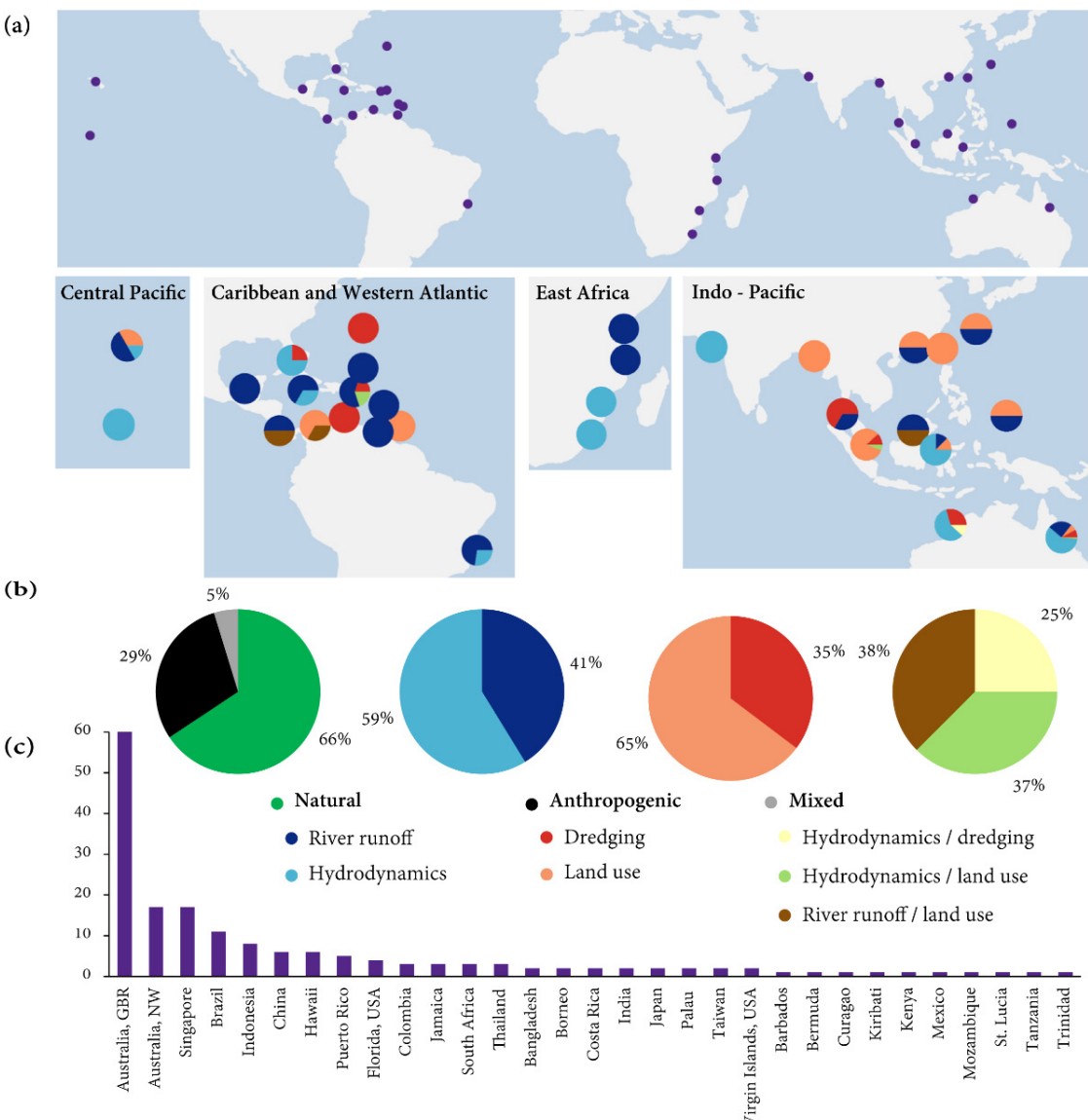

**Figure 3.** (**a**) Global distribution of the reviewed studies on turbid coral reefs. Colors in pie charts indicate the turbidity source described in the study. (**b**) Global percentage of turbidity source of studies conducted in natural (hydrodynamics *n* = 67, river runoff *n* = 47), anthropogenic (dredging *n* = 18, land use *n* = 33) and mixed (hydrodynamics/dredging *n* = 2, hydrodynamics/land use *n* = 3, river runoff/land use *n* = 3) environments. (**c**) Number of papers published in each location (*n* = 31). See Table S1 for details of studies and link to references.

## 3. Defining a Turbid Reef

Coral sediment thresholds are often poorly defined due to difficulties in accurately assessing sediment dynamics, which are influenced by sediment characteristics (e.g., size, shape, density), water properties (e.g., temperature, salinity), reef geomorphology (e.g., bathymetry) and the hydrodynamic regime (e.g., tidal range, wave energy, current velocity), and are highly variable across time and space [56,57]. Consequently, definitions of turbid reefs to date have largely focused on reef setting (e.g., inshore, sheltered, shallow, distance from rivers, and/or general observational data) as opposed to quantified levels of sedimentation and turbidity across a reef. Incorporating field sediment dynamics data (i.e., frequency, duration and severity of turbidity) is, however, an essential step forward in better defining turbid reefs and, more importantly, establishing a baseline that can be used to monitor changes in reef health in response to sediment exposure.

A quantified definition of turbid reefs requires high temporal (e.g., daily) and spatial (e.g., sites per reef) resolution of turbidity levels and sedimentation rates collected over prolonged periods of time (months-years) from several reefs. Further, a standardized framework of methods (i.e., measurement units, data logging frequency, principles for in situ instruments/traps placement, and troubleshooting guidelines) is required to improve our ability to compare data among sites and studies. Yet, only 7.7% of studies reviewed here report turbidity (or suspended sediment) levels and 18% report sedimentation rates. Of those that do include a sediment parameter (e.g., turbidity, suspended sediment, sedimentation), the length of time for data collection was usually 1–6 months, with the longest study being 4 years [58]. Furthermore, units for turbidity and sedimentation data vary. For turbidity, the most commonly used measurement unit is nephelometric turbidity unit (NTU) or formazin turbidity unit (FTU) when using turbidity loggers [20,40,59], with other studies focusing on light attenuation (as a proxy for turbidity) measured by light loggers (e.g., kd490, PAR, LUX) [11,60,61] or by Secchi disk [62–64]. Sedimentation rates are largely assessed using sediment traps (e.g., g m$^2$ d$^{-1}$) [65], but how they are deployed (e.g., size, height above the seabed, sampling intervals) considerably influences the interpretation of the data [65].

Due to limited (and incompatible) data on sediment regimes for turbid reefs, we were unable to constrain sediment exposure thresholds to define a turbid reef. Instead, we have identified three conceptual sediment exposure regimes: (1) persistent, (2) fluctuating, and (3) transitional (Figure 1, Table S1). Persistent turbid reefs are exposed to daily sustained suspended sediment loads above 5 mg L$^{-1}$/15 NTU (e.g., PSRC) [19]. Their coral community is likely dominated by sediment-tolerant corals (e.g., *Montipora*, *Turbinaria*, *Goniopora*, *Porites*, *Galaxea*, *Millepora*, *Montastraea*) [53,66,67] that typically inhabit high-energy settings (wind-wave and tidal currents), and have a high terrigenous sediment composition in their reef matrix [68,69]. Reefs considered to be fluctuating are exposed to episodic (daily to monthly) severe suspended sediment (>50 mg L$^{-1}$/>150 NTU) events interspersed by periods of often low turbidity or clear water (<5 mg L$^{-1}$/<15 NTU) (e.g., Marino Ballena, Pacific Costa Rica [70]; Pilbara, Western Australia [71]; Borneo [18]). Here, the coral community is also dominated by sediment-tolerant corals that inhabit mixed energy settings and have some (but less than persistent) terrigenous sediments in their reef matrix. Transitional reefs exhibit a sustained or stepped increase in turbidity over time (annual to decadal), often from clear water, or low turbidity, towards high turbidity levels (e.g., Singapore) [41]. These reefs have experienced a recent (<100 years) increase in sediment exposure and will likely show evidence of reef depth compression and loss in coral cover and diversity [40], although the extent of this will depend on the original baseline and the rate of change in sediment exposure.

## 4. The Past—Holocene Paleoecological Reconstructions of Turbid Coral Reefs

To address whether turbid reefs initiated and developed under natural turbid conditions, or if they have experienced recent anthropogenic-driven declines in water quality, analysis of the paleo-coral community composition and sedimentary facies spanning the growth history of the reef is needed [72]. Where the timing of past ecological transitions is known, we can then compare paleoecological records to shifts in paleoclimate and historical anthropogenic inputs as a means of assessing how these processes may have influenced the timing and nature of reef development, both in terms of reef accretion rates and coral community structure.

A total of 35 published studies have investigated the paleoecological record and/or growth history of turbid reefs with researchers framing questions around three broad themes: (1) the influence of natural drivers, such as regional sea-level oscillations, climate and cyclones on past turbid reef growth and present-day geomorphology [73]; (2) assessing the growth history of reefs in relation to natural and anthropogenic disturbances [72]; and (3) how the growth history and coral community structure of turbid reefs compare to nearby clear-water reefs [74]. Of the 35 published studies, 28 were located in Australia, with 25 of

these from the GBR, and the remaining three studies located in the Kimberley and Pilbara regions of Western Australia (Table S1). The seven studies located outside of Australia include single sites in the South (China) and East China Sea (Japan) [75,76], Espirito Santo in Brazil [77], Golfo Dulce in Costa Rica [78] and Phuket Thailand [79]. The key dataset common to all these studies is the recovery of a vertical reef framework through percussion and/or rotary drill cores. Both methods are limited to intertidal/shallow subtidal sections of the reef (reef flats) where researchers can operate the equipment subaerially during a low tide window. This has often limited the scope of reef coring campaigns to those reef habitats that are more easily accessible to researchers and still allow for potential recovery of the entire vertical reef growth history. Using hydraulic powered percussion coring methods, reef cores of up to 6.5 m in length have been recovered, recovering timeframes of ~7000 years [25]. While several studies have cored submerged reef slopes using manual percussion methods with scuba equipment, they have recovered cores of up to 4.5 m in length, which yielded narrower time windows (<200 years) of reef growth [72].

　　　Although reef cores can provide a continuous temporal record of reef growth, the width of the core barrel, typically between 75 and 100 mm in diameter, means that the spatial horizon is vastly underrepresented, and therefore limits a broader paleoecological examination of spatially contemporaneous coral communities through time. Despite this, there are several sedimentary (ratio of terrigenous vs carbonate sediments) [10], paleoecological (clear-water vs turbid-water coral species; coral death assemblages; foraminiferal assemblages) [72,74,80], taphonomic (e.g., style and nature of endolithic borers) [81], and geochemical indicators (e.g., stable isotopes) [82] that when combined with detailed chronostratigraphic analysis can provide information on the local paleo-environments, water quality, climate and coral community structure. For example, increasing suspended sediment load is a key indicator for a change in water quality, typically represented by the relative proportions of carbonate sediments to siliciclastic silts and clays contained within the reef matrix [83]. In the Kimberley, NW Australia, reef cores typically show a uniformly high ratio of siliciclastic to carbonate matrix sediments [84], suggesting that these reefs have adapted to, and developed under, a long-term turbidity regime. The ratio of siliciclastic to carbonate matrix sediments in nearshore GBR reef cores have either been dominated by siliciclastic sediments (e.g., Paluma Shoals) indicative of a long-term turbidity regime that is independent of any post-European degradation in coastal catchments [26], or characterized by more carbonate components up core as the reef shallows and move away from the seafloor/resuspension zone [85]. No study from contemporary turbid reefs on the GBR has shown evidence of a clear transition from carbonate-dominated to siliciclastic-dominated reef matrix sediments up core, which would suggest a shift in terrigenous sediment delivery to the coast. However, a study by Roff et al. (2013) did provide paleoecological evidence of coral community structural change from *Acropora*-dominated communities that transitioned to *Pavona*-dominated communities following European settlement, suggesting higher sediment and nutrient deposition to the reef [80]. Still, it should also be noted that as part of the natural evolution of turbid reefs, the supporting ecological communities do change as the reef vertically grows away from the seabed resuspension zone, reaches sea level and becomes depth constrained.

　　　Benthic foraminifera assemblages contained within reef matrix sediments can also provide additional information in support of paleo-environmental interpretations [80]. For example, Lewis et al. (2012) used the relative ratios of four foraminiferal species *Elphidium*, *Peneroplis*, *Amphistegina* and *Operculina* to provide insights into environmental conditions on fringing reefs including relative changes in water depth and turbidity [86], while Johnson et al. (2019) observed changes in foraminiferal assemblages from PSRC resulting from changes in hydrodynamic energy and light availability as the reef shallows towards sea-level [80], supporting similar depth-related transitions in coral community structure.

　　　The extensive reef coring campaigns on the GBR and in the Kimberley have revealed, through a range of sedimentological and paleoecological indicators, that present-day turbid-reef coral communities within these regions are experiencing a turbidity regime

that has persisted for much of their reef growth history. Still, there are many regions where turbid reefs have been reported (Figure 3), particularly throughout Southeast Asia, which have seen significant increases in coastal populations and land use change, and with these, uncertainty around the extent to which their inshore coral communities have transitioned towards turbid ecology in response to decreasing water quality [55,87,88]. The poor global representation in understanding turbid reefs' Holocene growth is a knowledge gap that requires further reef coring efforts in these regions, combining paleoecological reconstructions with environmental proxies in order to establish the baseline shift of these nearshore reefs from their original state and future trajectories.

## 5. The Present (1900 to Present Day)

There has been a recent increase in the number of publications on turbid reefs, with 82% of the papers reviewed here published since 2000, reflecting an increased awareness of these reef types and their potential value. Here we compiled the global distribution of published studies on turbid reefs, and discuss different turbidity sources and environmental settings. Further, we compare the two most well-studied turbid-reef systems, persistent-natural (PSRC) and transitional-anthropogenic (Singapore reefs), by exploring differences in their turbidity status, current ecological state (coral cover, community structure, accretion rate) and environmental conditions.

### 5.1. Global Distribution, Sources of Turbidity and Environmental Setting

A total of 31 turbid-reef systems were found in this review and are globally distributed through coastal waters (Figure 3). Natural turbid reefs constitute 66% ($n = 114$) of the reviewed studies and are found worldwide, 29% ($n = 51$) are anthropogenic turbid reefs and 5% ($n = 8$) of all studies reported mixed sources of turbidity (Figure 3).

Of the 173 papers, 45.6% were on turbid reefs in Australia ($n = 79$), of which 78.4% ($n = 62$) were on the inshore GBR where a strong southerly wind regime drives local wave resuspension [8,89]. In contrast, only 17 studies have focused on NW Australian reefs, and of these, 41% ($n = 7$) relate to the impact of dredging activities on reefs in the Pilbara region, 30% focused on the Kimberley reefs' geological record and only 29% on coral reef ecology and/or physiology throughout Barrow Island [90–93] and the Dampier Archipelago [94,95]. The Kimberley region has a combined reef area of almost 2000 km$^2$ [96], and while these reefs are exposed to a high turbidity regime due to large tidal ranges (>11 m) and associated tidal currents, there are only nine publications from this region on turbid reefs [25,90,97–99]. The lack of studies is most likely due to the remote location and lack of research infrastructure, which makes them logistically challenging to access.

Major sources of turbidity were found to be region-specific. For example, the four (2.3%) studies conducted in East Africa are all classified as natural turbid reefs. In Kenya [100] and Tanzania, a biodiversity hotspot in the Western Indian Ocean [101], turbidity was attributed to terrestrial sediment input from river runoff, while in Mozambique [102] and South Africa [103,104] turbidity was driven by the regional hydrodynamics. In the Caribbean and western Atlantic, in countries such as Jamaica [105,106], Costa Rica [70] and Mexico [107], the major driver for turbidity was also natural river runoff (53%), although many of the studies ($n = 11$, 34%) from this region were conducted at the Abrolhos Bank, Brazil [30,60,67,108] located offshore of the Amazon river [109]. The most extreme mixed (natural-river runoff/anthropogenic-land use) turbid environment in this region is found in Cartagena Bay, Colombia, where turbidity surrounding Varadero Reef (~45% coral cover) [16], situated < 12 km from Cartagena city (>1 million people) is largely related to coastal development, industrial and sewage waste, and sediment discharge (144 × 10$^6$ tons of suspended solids per year) from the Magdalen River [15,110,111]. In the Indo-Pacific region (30% of studies; not including Australia), high turbidity was largely attributed to anthropogenic sources (49% land use, 7% dredging). For example, in studies from Singapore, where most research on turbid reefs in this region have been conducted ($n = 17$; 32% of the Indo-Pacific), 82% report changes in land use as the main source of

turbidity [62,112,113], and the remaining 18% (*n* = 3) refer to dredging [114,115] and mixed hydrodynamics/land use [116].

Globally, reefs that have initiated and developed within natural turbid conditions are typically found in one of six environmental settings (see Figure 4 in [53]). These include (1) wave protected (e.g., the leeward side of submerged rocky outcrops (e.g., Abrolhos Islands, Brazil [58]; Sodwana Bay, South Africa [66]), (2) open coast, sedimentary shore-lines (e.g., Paluma Shoals, situated on intertidal terrigenous sand/mud, central GBR [117]), (3) offshore terrigenous shelves (e.g., Inhaca Island, southern Mozambique [118]), (4) fluvial embayment (e.g., Rio Bueno, Jamaica [106]), (5) river deltas (e.g., Bay of Baten, Indonesia [119]; Magdalena River, Colombia [120]; Pearl River, Hong Kong [16]), and (6) muddy coastal embayments (e.g., Phuket, South Thailand [79]; Talbot Bay, Kimberley, Western Australia [17,84]). These different sedimentary and geomorphic settings highlight the broad range of natural environmental conditions where turbid reefs have initiated and developed and could be used to distinguish natural turbid reefs (persistent, fluctuating) from those reefs that have transitioned to turbid (or to more turbid) during the Anthropocene.

*5.2. Paluma Shoals Reef Complex, Great Barrier Reef, Australia—Natural (Persistent) Turbid Reef*

Paluma Shoals Reef Complex (PSRC), located in the shallow waters (<20 m) of Halifax Bay, central GBR, Australia (19°6′52.2″ S, 146°32′58.92″ E), is relatively remote, with the nearest major urban development (Townsville with 195,084 people in 2020) ~30 km to the south [26,121,122]. This turbid nearshore shoal comprises seven disconnected fringing reef structures [14]. The two shore-attached reefs emerge under the lowest astronomical tide (LAT) while the offshore structures are fully submerged [7,8,14,26]. The persistent turbidity at PSRC is the result of wind-driven waves and tidal resuspension processes (tidal range: 3.6 m), with high-turbidity events that can reach up to 175 NTU [8,89,122]. Sedimentation rates on shore-attached reefs differ depending on reef geomorphological location, with 0.9 g m$^2$ d$^{-1}$ on the reef flat and 120 g m$^2$ d$^{-1}$ in sheltered leeward locations (Table 1) [53].

PSRC is a geologically young reef with initiation dates ranging from ~700–2000 calibrated years before present (cal. y BP) [122,123]. Periods of rapid reef growth (7.8 mm year$^{-1}$) have occurred under turbid conditions [122], and have been attributed to the incorporation of terrestrial sediment into the matrix [10,26,117,123]. Reef core records indicate a constant coral community (Table 1), for at least the past millennium, which exhibits no evidence of community shifts associated with post-European settlement (ca. 1850 AD) [19,21,26]. These data suggest that PSRC is a persistent naturally turbid reef, with a stable coral community.

Naturally high turbidity and associated light attenuation in Halifax Bay confines reef-building corals to a shallow zone of ~4 m below LAT [123]. Still, PSRC structural complexity and average coral cover is high (~38%) [121], as is the rate of net carbonate production (6.9 ± 10 kg m$^{-2}$ year$^{-1}$) and net vertical accretion (average = 2.97 mm year$^{-1}$, maximum = 6.4 mm year$^{-1}$) [19], demonstrating rapid reef-building potential under high turbidity [122,123]. Elevated above the seafloor, the PSRC coral community comprises structurally complex, fast-growing taxa (e.g., *Montipora* spp., *Turbinaria* spp., *Acropora* sp.) that feed both autotrophically and heterotrophically [123,124]. Closer to the seafloor, mostly sediment-tolerant, heterotrophic coral taxa are found (e.g., *Galaxea* sp., *Lobophyllia* sp., *Euphyllia* sp.) [14,124].

Several heatwave events in the past decade have caused severe coral bleaching events globally and on the GBR [125,126]. After the unprecedented 2015–2016 event, Morgan et al. (2017) found that PSRC coral colonies exhibited high tolerance to bleaching with no significant declines in coral cover (pre-warming: 48 ± 20%; post-warming: 55 ± 26%) or changes in coral community structure [7], while several offshore reefs in northern and central GBR exhibited high bleaching severity of 50–100% of the coral community [127]. Furthermore, responses of specific taxa to the warm water event were in contrast to their clear-water counterparts. For example, *Acropora* corals, which are known to be highly susceptible to bleaching on clear-water reefs [128], were the least impacted of the coral

species present at the PSRC, a phenomenon that has been observed within other turbid settings [7,18].

### 5.3. The Southern Islands Group, Singapore—Anthropogenic (Transitional) Turbid Reef

Singapore, located in Southeast Asia at the edge of the Coral Triangle, is home to the world's busiest port with ~500 large commercial vessels passing through every month [55,129,130]. Since 1965, Singapore has expanded its island area by 25% through extensive reclamation projects [131,132] as a means of accommodating the rapidly growing population (5.69 million people in 2020) and industry [41,133,134]. This has resulted in a dramatic transformation of Singapore's seascape and shoreline, as well as a 60% reduction in coral reef area [131,134].

Information on coral reef cover and composition pre-1960s is limited to anecdotal observations and historical records. For example, Crawfurd (1830) described the superior beauty of the numerous southern offshore islands where most of the coral reefs were located when he sailed through in 1822 [135]. Using historical maps, Hilton and Manning (1995) estimated that the total area of intertidal reefs in Singapore was ~32.2 km$^2$ in 1922 with corals growing down to 10 m depth [136]. Long-term ecological monitoring [137] since the 1980s estimates that Singapore's reefs previously supported ~250 species of scleractinian coral, out of which about 160 species are locally extant to date (Table 1) [31,55,87].

Today, Singapore's coral reefs form compact fringing and shallow patch reefs [31,138]. Due to high baseline turbidity (4.8–6.6 NTU) [139], which limits light penetration, coral growth rarely extends beyond 6 m depth [40] and land reclamation has reduced reef flats [62]. High turbidity and sedimentation rates (5 to 35 mg cm$^{-2}$ d$^{-1}$) [139] have most likely influenced the coral community structure, which is dominated by foliose, laminar and sub-massive taxa, and few fast-growing tabular and branching acroporid corals at sites furthest offshore [137]. Consequently, current average rates of vertical reef accretion, calculated using carbonate budgets, are estimated at 0.35–2.76 mm year$^{-1}$ [39]. This suggests that these reefs are currently in a state of limited reef growth. Still, diverse coral communities exist and coral cover is high (13–49%) with many sites above the current average (~25%) for the Indo-Pacific [113,134,140].

Singapore reefs' turbidity has increased over the past 30 years [134,138,141]. The current lack of published paleoecological data on the reefs, however, reduces our capacity to assess the timeframe over which terrigenous sediments have influenced the coral community and reef development as well as when reef development initiated. The reduction in coral cover, changes to the coral community structure and evidence of coral growth zone shrinking [137] suggest a transitioning anthropogenic reef, although we cannot conclude whether these reefs were clear-water or naturally turbid prior to anthropogenic disturbance.

Although major acute disturbances present on other Indo-Pacific reefs, such as crown-of-thorns starfish or cyclonic storms [141,142], are absent in Singapore, they have experienced two major bleaching events, one in 1998 and one in 2010 [137,143]. During the 2010 bleaching event, ~60% of colonies were moderately or severely bleached, but only 5–30% of colonies completely bleached with <10% mortality reported [137,144]. The rapid recovery recorded on Singapore's reefs is attributed to the stress-tolerant, slower growing (e.g., *Porites* and *Platygyra*) and generalist coral taxa (e.g., *Merulina*) that dominate the coral community [63,137]. These coral taxa are also considered to be more resilient to future predicted increases in ocean warming [39,54,55].

### 5.4. PSRC vs. Singapore

PSRC and Singapore represent two different types of turbid reefs, natural and anthropogenic respectively.

The contrasting turbidity regimes in Singapore reefs and PSRC may, in part, explain important differences in coral coverage and carbonate production rates in various types of turbid reefs. In Singapore, turbidity levels are lower than in PSRC, but reefs experience higher sedimentation rates [53,139,152]. Frequent exposure to wind-driven waves at PSRC

leads to elevated sediment resuspension and near persistent turbidity, whereas in Singapore, the sheltered tidal-controlled system is characterized by low energy and therefore, higher levels of sedimentation [40]. Sedimentation is considered to be more detrimental to coral settlement, growth and survival than lower light levels [153–155], potentially resulting in lower coral coverage. As such, the balance between sediment settling and resuspension is as important as the volume of sediments entering the nearshore environment.

**Table 1.** Comparison of environmental, physical and ecological parameters at Paluma Shoals Reef Complex (PSRC) and Singapore.

| | | Paluma Shoals Reef Complex | Singapore |
|---|---|---|---|
| **Nearest urban development** | | Townsville ~30 km, 195,084 people (in 2020) [145] | Singapore < 6 km, 5.69 million (in 2020) [133] |
| **Reef initiation period** | | 1700–1000 YBP [26,117,121] | No data available |
| **Stressors** | **Global** | Cyclones, heat waves, crown-of-thorns starfish [14,142,146,147] | Heat waves |
| | **Local** | N/A | Dredging, coastal development, ship traffic [138,143] |
| **Sea surface temperature (°C)** | | 25–28 [148] | 27–31 [134,149,150] |
| **Turbidity regime** | | Natural-persistent (wind-waves, tidal currents, river plumes) [8,89,123] | Anthropogenic-transitional (dredging, coastal development) [41,131,132] |
| **Turbidity (NTU)** | | 15–50 [8,10,19,40] | 4.8–6.6 [149,151] |
| **Sedimentation rate (average) [1]** | | 60.5 g m$^2$ d$^{-1}$ [53] | 176 g m$^2$ d$^{-1}$ [149] |
| **Coral genera [2]** | | *Montipora (50%)*, *Acropora (15%)*, *Turbinaria (12%)*, *Porites (1.5%)*, *Lobophyllia*, *Stylophora*, *Seriatopora*, *Pavona*, *Goniastrea*, *Favia*, *Favites Platygyra*, *Goniopora*, *Galaxea*, *Psammocora*, *Cyphastrea*, *Hydnophora*, *Symphyllia*, *Echinopora*, *Pachyseris*, *Alveopora*, *Fungia*, *Euphyllia* [7] | *Pectinia (11–19%)*, *Pachyseris (7–14%)*, *Merulina (6–12%)*, *Montipora (7%)*, *Porites (6%)*, *Echinopora (4%)*, *Platygyra (4%)*, *Acropora*, *Pocillopora*, *Pavona*, *Goniastrea*, *Favia*, *Favites*, *Lobophyllia*, *Goniopora*, *Galaxea*, *Montastraea*, *Diploastrea*, *Cyphastrea*, *Hydnophora*, *Symphyllia*, *Echinophyllia*, *Oxypora*, *Leptoseris*, *Leptastrea*, *Fungia* [87,137,150] |
| **Coral cover (average)** | | 38% [14,19] | 31% [113,140] |
| **Reef geomorphology** | | Fringing (inner-shelf, coastal reefs) and offshore patch reefs [10] | Fringing or patch reefs near the southern islands [31] |
| **Coral growth depth range** | | <6 m [123] | <6 m [62] |
| **Reef area** | | ~16 km$^2$ [26] | ~9.5 km$^2$ [131] |
| **Carbonate budget (CaCO$_3$)** | | ~6.9 kg m$^2$ year$^{-1}$ [19] | ~3.7 kg m$^2$ year$^{-1}$ [39] |
| **Reef accretion potential (average, based on carbonate budget values)** | | 2.97 mm year$^{-1}$ [19] | 1.55 mm year$^{-1}$ [74] |

[1] In PSRC measured as net sedimentation using sediment trays and in Singapore measured as gross sediment accumulation using sediment traps. [2] Coral genera (%) is percentage cover at that site. Species without (%) are <1% of coral cover. Bold genera are found at both reefs, underlined genera are considered sediment tolerant. All data shown in the table were acquired from and belong to the referenced publications.

In the coastal waters of Southeast Asia, including Singapore, the high presence of terrestrial derived dissolved organic matter contributes to low light availability that further compounds suspended sediment impacts [43]. In contrast, PSRC, although located in the wet Australian tropics, exhibits lower nutrient levels due to its remoteness from an urban center along with effective regulations and management of water catchments in the area [5]. Thus, the dominant source of turbidity is suspended sediment.

Despite differences in the turbidity regime (length of exposure and source), both reef systems are largely composed of sediment-tolerant species (Table 1) that have also demonstrated resilience to warm water temperatures. In Singapore, it has been suggested that given its historical record of warmer waters, the coral community has adapted and is more tolerant to elevated SST [137]. PSRC does not have the same history of exposure to warm waters, which suggests that resilience may partially be explained by a turbidity-driven reduction in UV, which acts as a synergistic stressor decreasing rates of bleaching [156,157].

The differences in reef setting and turbidity regime together with differences in reef ecology, functionality (e.g., carbonate production) and potential resilience support the acknowledgment that natural and anthropogenic turbid reefs are two different reef types. This distinction is particularly important when assessing their value for future reef conservation plans.

## 6. The Future—Facing Local and Global Stressors

Intensifying human population pressure and land use change associated with coastal development will increase sediment loads within tropical coastal waters [40,158,159]. This may expand the range of turbid coral habitat [160,161], as well as increase turbidity and sedimentation levels on existing turbid reefs. Eutrophication of coastal waters, which is often closely associated with high terrestrial sediment inputs, also contributes to turbidity and presents an additional serious threat (e.g., increased bioerosion) to all coastal reefs [162,163], but particularly to reefs located near urban centers where nutrient loading is greatest [164,165]. Reef resilience to global climate-related impacts (e.g., warming oceans, cyclones, rising sea levels) will be influenced by the coral communities' ability to cope with these local threats. Yet, we have a limited understanding of how these multiple stressors interact with water quality to influence reef function. This, therefore, limits our capacity to confidently identify if turbid reefs may have resilience to future threats. Here we review the current knowledge regarding major future threats to turbid reefs and how they may respond to localized and global stressors, identify potential indicators of resilience and outline future research avenues for turbid coral reefs (Table 2).

**Table 2.** Summary of major threats to turbid reefs, potential attributes of resilience and outstanding research questions.

| Threat | Resilience Attributes | Outstanding Questions |
|---|---|---|
| **Increasing sediment loads** | Sediment-tolerant corals (e.g., morphological adaptation, enhanced photo-acclimatization to low light, heterotrophic feeding) | What are the molecular components that improve a coral's ability to grow, adapt and acclimate to turbid conditions? |
| | Higher energy hydrodynamic setting | Is there a threshold energy level that is more likely to support turbid reef growth and development? |
| **Eutrophication** | Remote settings (e.g., >50 km from urban areas) | How do nutrient inputs influence coral growth and skeletogenesis, and what are the consequences for longer-term reef development? How will bioerosion intensity change with increased eutrophication? |
| | Effective conservation, management and regulation plan | What is the coral community threshold to nutrient input? What are the best ways to control nutrient flow into coastal catchments? |
| **Warming oceans** | Persistent turbid reefs where corals have adapted to low light and where suspended sediments may reduce stress from UV radiation | What is the relationship between suspended sediment concentrations and reduced stress from UV (during bleaching events)? |
| | A higher proportion of heterotrophic corals that can utilize this energy resource during bleaching events | By how much does heterotrophy extend the survival rate of bleached corals and improve recovery rates? |
| | Heat-tolerant symbionts | How do survival and recovery rates differ among different coral/symbiont clade associations? |
| **Storm severity** | Higher skeletal density | To what extent does lower coral skeletal density influence mechanical damage during a storm event? |
| | Massive and encrusting corals reef communities-dominated reef | How does the ratio of branching to encrusting to massive influence rates of coral dislodgement (with cyclone energy)? What has more influence on rates of coral dislodgement during storm events: coral community structure or substrate strength? |
| **Ocean acidification** | Unknown | How do turbidity and/or sedimentation affect coral physiology under different OA scenarios? |
| **Sea-level rise** | Higher net carbonate production | What is the vertical growth potential (i.e., carbonate budgets) of present day turbid coral communities? |
| | The reef structure is at/or close to sea level | What are the SLR projections for tropical coastal settings where most of the turbid reefs are located? |
| | | Will corals be able to colonize algal/sediment substrates as accommodation space above reefs increase? |
| | | How will SLR change turbidity conditions and sedimentation on reefs? |

### 6.1. High Sediment Loads

Turbid-reef corals can thrive under high sediment loads due to a combination of acclimation and adaptation mechanisms. Acclimation mechanisms include increases in photo-efficiency in response to low light [114,166], heterotrophy to offset reduced photosynthetic energy production [124], and mucus production to reduce sedimentation effects [167,168]. Previous studies have demonstrated that turbid water corals can rapidly acclimate (hours to days) to sudden spikes in suspended sediments. For example, Browne et al. (2014) found that *Platygyra sinensis* was able to increase its photosynthetic yield by 12% (0.58 to 0.65) following a 90-minute exposure to a high-turbidity event (242 mg $L^{-1}$, 13 mg $cm^2$ $h^{-1}$) [169]. Therefore, corals that can acclimate quickly to rapid declines in light are likely to dominate turbid reefs [20]. Likewise, corals with morphological adaptations to highly variable environments also tend to outcompete other coral species [103]. For example, *Turbinaria* spp. is often considered a turbid-water coral [170] that tends to grow vertically in turbid environments to form a cone shape, thereby reducing coral surface area for sedimentation [171]. More recent studies investigating proteomes have found that corals growing in turbid waters have also adapted at the molecular level by upregulating detox-proteins and those involved in immune responses, which was suggested to provide these corals with elevated resilience to poor water quality [172,173]. The identification of molecular markers that potentially provide the coral with the ability to better cope with high sediment and nutrient loads is a promising avenue for future research.

Furthermore, there is evidence that the negative impacts of turbidity on coral physiology are less than those of sedimentation. A review of coral responses to turbidity found that stress (e.g., reduced growth, bleaching, mortality) was not commonly observed until corals were exposed to turbidity over 150 mg $L^{-1}$ for a duration of several weeks [149]. In contrast, signs of sedimentation stress (e.g., tissue necrosis) were observed within days. As such, reefs that are dominated by corals that are better able to cope with sediments through acclimation responses and/or adaptive features (e.g., morphology, sacrificial zones) and are located in higher energy hydrodynamic settings where sediments are more frequently resuspended and removed may be more resilient to future increases in sediment loads.

### 6.2. Eutrophication

Localized drivers of future increases in sediment delivery are expected to increase nutrient concentrations in coastal regions [174,175]. Over evolutionary timescales, corals adapted to oligotrophic waters through the establishment of the symbiotic association with the photosynthetic dinoflagellate algae Symbiodiniaceae [176,177]. Despite contradictory reports on the impact of nutrients on corals, most studies suggest that high nutrient levels will be detrimental to coral reefs [162,178,179]. For example, a comprehensive in situ study of elevated nutrient effects on the reef corals at One Tree Island, Australia, found that high nutrient levels resulted in lower coral skeletal density and lower reproductive potential [180]. A review by D'Angelo and Wiedenmann (2014) highlighted several direct and indirect nutrient pathways that can negatively impact coral physiology and ecosystem function, and emphasized the importance of phytoplankton blooms in converting increased nutrient levels to nutrient stress on coral reefs, even to those far from the primary source of nutrient enrichment [179]. In addition, there is growing evidence to suggest that reefs exposed to nutrients are more susceptible to bleaching [181–183]. Contradicting studies such as from Sawall et al. (2011) in Sulawesi suggested that some corals (e.g., *Stylophora* spp.) may benefit from eutrophication through increased heterotrophy, which then provides the energy for increased mucus production and sediment clearing [184]. Given that the effects of nutrients vary among coral species (due to differences in acclimation/adaptation potential) and reef sites (due to synergistic effects with other environmental stressors), more work is needed to identify coral characteristics or species, and/or reef characteristics (e.g., higher energy) that increase resilience to elevated nutrients.

High levels of nutrients in coastal waters can also indirectly influence corals by increasing the abundance of other reef organisms that compete with corals for space. For

example, increase in algal cover reduces suitable substrate available for coral recruitment, shades coral colonies [185], and can enhance the prevalence of coral diseases, which in turn reduces coral function and elevates rates of coral mortality [186]. Heterotrophic bioeroders (e.g., sponges) that bore into the reef framework, weakening the reef structure can also increase in abundance [187]. Therefore, turbid reefs situated in an urbanized setting may be more at risk from future global stressors than turbid reefs in remote settings.

### 6.3. Warming Oceans

Recent models and field-based evidence support the hypothesis that corals in turbid waters are more resilient to prolonged periods of heat stress that typically result in mass coral bleaching events. This evidence largely comes from field observations during ocean warming events where turbid reefs have demonstrated lower levels of bleaching and mortality than their clear-water counterparts, despite comparable SSTs [6,7,18,113]. Although the mechanism/s that provide the increase in resilience to warmer temperatures is not fully understood [188], it is likely due to either one or a combination of: (1) suspended sediments that reduce stress from UV radiation, which is known to increase susceptibility to warmer temperatures [3,99,156], (2) suspended sediments and associated nutrients provide an additional energy source for corals via heterotrophic feeding potentially negating the energy deficits from reduced light and photosynthesis [189,190], and (3) corals in shallow turbid waters exposed to more variable temperature regimes have established a symbiosis with more heat-tolerant Symbiodiniaceae clades [191]. Future research should seek to confirm if these field observations can be repeated ex situ to determine temperature thresholds with turbidity levels, and quantitatively assess the importance of these potential mechanisms that confer bleaching resilience. These data could then potentially be harnessed as a means of transferring resilience to clear-water reefs.

### 6.4. Increased Storm Severity and Ocean Acidification

Despite evidence that turbidity may provide some resilience to warmer waters, the relative impact of other climate change outcomes such as increased storm severity and ocean acidification is likely greater on turbid reefs. Several studies have demonstrated that coral skeletal density is lower on turbid reefs than on clear-water reefs [192,193] due to a trade-off with higher linear extension rates driven by limited light availability [107,194]. Lower skeletal density in turbid-water corals can increase susceptibility to breakage during storm events and cyclones, resulting in lower coral cover and reduced habitat complexity [193,195]. Ocean acidification will reduce net carbonate production as it changes the chemical components of the water, making it harder for coral and other calcifying organisms to build their calcium carbonate skeleton [159,196]. There is currently no data to suggest that turbid reefs are more or less vulnerable to the effects of ocean acidification (OA) than clear-water reefs; however, recent work by Mollica et al. (2018) indicates that OA negatively influences skeletal density and not linear extension rates [197]. Hence, the low skeletal densities already observed on turbid reefs could be further reduced, making them even more susceptible to breakage, thereby having implications for reef accretionary potential.

### 6.5. Sea-Level Rise

The relative water depth above coral reefs as sea levels rise (SLR) will arguably have the greatest impact on turbid coral communities. This is because surface light is attenuated more rapidly with increasing turbidity [198], and as a result, turbid reefs exhibit a shallow photic zone (<12 m) that limits the maximum depth range of coral growth [26,123], known as vertical reef compression [40,55]. Recent modeling projections of turbid reef morphology and habitat change under future SLR scenarios (RCP4.5 and RCP 8.5), utilizing combined reef core records and ecological datasets, demonstrated that shifts in the spatial extent of benthic communities may be disproportionate to the absolute changes in relative water depth above reefs [40]. Present-day reef morphology and surrounding seafloor depth of turbid reefs play a key role in future coral habitat by influencing local environmental

conditions (e.g., wave exposure, emergence time, sediment resuspension, light availability) as sea levels rise. For example, shallow reef flat environments, which are presently sea-level constrained and comprise lower coral cover, may 'turn on' carbonate productivity through the establishment of a complex reef framework [23]. In contrast, deeper reef-slope corals may increasingly move below the euphotic depth, and higher sedimentation may convert the benthos to soft-sediment cover [40,137]. Successful transitions from reef flat environments to higher coral cover states is reliant on coral recruitment to sediment-bound algal turf substrates [153].

Changes to coral habitat will not only influence reef biodiversity and their conservation status, but also future reef morphological development, as altered benthic communities modify reef accretion capacity [199,200]. Morgan et al. (2020) suggested that the magnitude and rate of habitat change on turbid reefs is linked to three main interacting factors: (1) regional rates of SLR, (2) vertical reef accretion capacity by coral communities, and (3) local turbidity regimes [122]. As a result, anthropogenic turbid (transitional/persistent) reefs (e.g., Singapore), which already experience extreme vertical reef compression and limited reef growth potential [150], are likely to be more impacted by SLR than natural turbid (fluctuating/persistent) reefs (e.g., PSRC), where background turbidity is lower and corals experience periods of high light exposure. Furthermore, SLR in an urbanized setting is likely to occur in synergy with continuing poor water quality that may cause further light attenuation and shoaling of the euphotic depth, exacerbating the effects of increases in water level [122]. Reef-scale sediment dynamics and turbidity may also change under a higher sea level, potentially reducing tidal current velocities across reefs in tidally-dominated settings (e.g., Singapore), and elevating suspended sediment concentration on reefs that experience higher wave exposure (e.g., PSRC) [158]. Indeed, these regional changes in hydrodynamics may also drive an expansion of turbid reefs as shorelines retreat, scouring fine sediment and altering nearshore bathymetry to establish new substrate for early colonizing coral taxa.

## 7. Conclusions

Turbid coral reefs are likely to increase in abundance with future climate change effects, such as sea-level rise and increasing storm and rainfall events, as well as from anthropogenic influences, including land use change and the expansion of urban centers. There has been a recent (<20 years) increase in research on these understudied reef systems (with exception of PSRC and Singapore), yet due to the use of inconsistent methods and poor spatiotemporal data collection, comprehensive accounts of the sedimentary regime (and other environmental parameters) are rare and typically incompatible. Consequently, identifying quantitative turbidity thresholds that can be used to define a turbid reef is not possible. Instead, we identified three turbidity regimes (persistent, fluctuating, transitional), which take into account environmental variability and timeframes, and highlight the importance of detecting the turbidity source (i.e., natural versus anthropogenic) as a means of better characterizing turbid reefs. By acknowledging important differences in turbidity regimes and sources among turbid reefs, we are better equipped to identify those that may be more resilient to future climate change and serve as conservation hotspots.

There are still many unknowns regarding how turbid reefs will respond to future global and local threats. Evidence from the recent geological past suggests that inshore turbid reefs on the GBR have not experienced a transition to a more siliciclastic-dominated reef matrix up core, or a shift in community composition, as a result of European settlement. In addition, there is growing evidence that these reefs are more resilient to bleaching events than clear-water reefs, although the mechanism/s that confer resilience warrant further investigation. Likewise, there is little information on how these reefs will respond to declining ocean pH and increased storm severity, although it is likely that given their shallow water setting and lower skeletal density, turbid reefs may be less resilient to these two threats. Some would argue that the regional rate of SLR is the key threat to the survival of turbid reefs given the higher rates of light attenuation with depth. However, until

we improve sea-level projections for tropical coastal settings, and quantify rates of net carbonate production and reef accretion potential, the impact of this threat is difficult to predict.

**Supplementary Materials:** The following are available online at https://www.mdpi.com/article/10.3390/d13060251/s1, Table S1: Turbid coral reefs global map references list.

**Author Contributions:** Conceptualization, A.Z., N.K.B. and M.O.; investigation, A.Z.; data curation, A.Z.; writing—original draft preparation A.Z., N.K.B., M.O. and K.M. All authors have read and agreed to the published version of the manuscript.

**Funding:** A.Z. is funded by the Holsworth Wildlife Research Endowment PG51006700 from the Ecological Society of Australia and supported, in kind, by the Minderoo Foundation through the Minderoo Foundation Exmouth Research Laboratory (MERL). A.Z. is an awardee of the Robson and Robertson award from the Oceans Institute, University of Western Australia. N.B. is funded by a DECRA Fellowship DE180100391 from Curtin University, Perth. K.M. is a beneficiary of an AXA Research Fund Postdoctoral Grant and a Nanyang Presidential Postdoctoral Fellowship.

**Institutional Review Board Statement:** Not applicable.

**Informed Consent Statement:** Not applicable.

**Acknowledgments:** A.Z. would like to acknowledge the Minderoo-Exmouth Research Laboratory (MERL) operated by the Minderoo Foundation and its staff for their in-kind support. We would like to thank Renae Hovey for reviewing the manuscript. Thank you to the REIF group at the University of Western Australia and Curtin University for their honest advice and continued support. A special thankyou to Chris Whitwell for his help with constructing the global map and to Belinda Martin from Ooid Scientific for her help with designing the concept figure.

**Conflicts of Interest:** The authors declare no conflict of interest.

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
