# Peer review of "Turbid Coral Reefs: Past, Present and Future—A Review"

_diversity, doi:10.3390/d13060251_

Round 1

Reviewer 1 Report

This is a comprehensive and well written review of turbid coral reefs, which synthesizes what is known about modern turbid reef and their Holocene foundations, and gives some suggestions about future of turbid reefs and future research of them.

A few minor changes will improve the paper (see attached copy with comments). Among them, I suggest clarifying that the “past” of turbid reefs dealt with in the paper refers only to the Holocene story of the cases in which drilling has been carried out. In this sense, I am not sure about the purpose of the list of turbid reefs in the supplementary material regarding fossil examples. Why, for example, the Miocene reefs studied by Wilson (2005) and Santodomingo et al. (2015) are included in the list but not the one of Novak et al. (2013) from the same area.

The modern reefs of East Kalimantan, although included in the supplementary list are not reflected in the global map.

Reviewer 2 Report

This is a good first attempt to organize the literature to date on turbid reefs and the impacts of turbidity on coral assemblages. The logic is sound and I have no substantive criticisms.

The authors might look at papers by Aronson and colleagues In Ecology and Ecological Monographs on submarine coring of lagoonal reefs in Belize and Caribbean Panama, as well papers by Toth et al. (Science 2012 and others) on submarine coring off the Pacific coast of Panama. Contrary to a statement in the present manuscript, those coring studies recovered millennia of reef-framework from what were likely persistently turbid environments.

Author Response

This is a good first attempt to organize the literature to date on turbid reefs and the impacts of turbidity on coral assemblages. The logic is sound and I have no substantive criticisms.

We thank the reviewer for their positive feedback.

The authors might look at papers by Aronson and colleagues In Ecology and Ecological Monographs on submarine coring of lagoonal reefs in Belize and Caribbean Panama, as well papers by Toth et al. (Science 2012 and others) on submarine coring off the Pacific coast of Panama. Contrary to a statement in the present manuscript, those coring studies recovered millennia of reef-framework from what were likely persistently turbid environments.

The papers suggested by the reviewer were not included in this manuscript as our literature review process was very conservative (see methods section). These sites in associated papers did not directly refer to turbidity as an influencing factor at their study sites, and as such, we excluded them from the review. The statement regarding reef-framework recovered cores is based on the coring studies included in the present manuscript.